# Antibacterial Activity of Phloretin Against *Vibrio parahaemolyticus* and Its Application in Seafood

**DOI:** 10.3390/foods13223537

**Published:** 2024-11-06

**Authors:** Siyang Chen, Wenxiu Zhu, Yiqun Zhan, Xiaodong Xia

**Affiliations:** State Key Laboratory of Marine Food Processing and Safety Control, National Engineering Research Center of Seafood, School of Food Science and Technology, Dalian Polytechnic University, Dalian 116034, China; csy10_4sym@163.com (S.C.); 18742058310@163.com (W.Z.); 2317010124@xy.dlpu.edu.cn (Y.Z.)

**Keywords:** *Vibrio parahaemolyticus* ATCC17802, phloretin, antimicrobial mechanism, enzyme activity, seafood

## Abstract

Although phloretin is widely utilized in the food industry as an additive, its effects on foodborne pathogens remain insufficiently investigated. This study aimed to evaluate the antimicrobial properties of phloretin (PHL) against *Vibrio parahaemolyticus* (*V. parahaemolyticus*) and to elucidate the potential mechanisms of action. After PHL treatment, alterations in the cell morphology, cell microstructure, and intracellular contents of *V. parahaemolyticus* were assessed. Scanning electron microscopy revealed substantial damage to cell integrity, subsequent to PHL treatment. A notable reduction in intracellular components, including proteins, ATP, and DNA, was observed in samples treated with PHL. PHL was shown to inhibit the activities of *ATPase*, β-galactosidase, and respiratory chain dehydrogenase in *V. parahaemolyticus*. Furthermore, it was demonstrated to elevate the intracellular levels of reactive oxygen species and promote cell death. After being applied to sea bass, shrimp, and oysters, PHL effectively inactivated *V. parahaemolyticus* in these seafoods. These findings demonstrate that PHL has potential for application in seafood to control *V. parahaemolyticus.*

## 1. Introduction

Seafood consumption is on the rise globally, which is attributed to its high nutritional benefits and palatability. Nonetheless, seafood is particularly vulnerable to contamination by various foodborne pathogens, posing substantial health risks to consumers [1]. The consumption of food tainted with foodborne pathogens commonly results in mild gastroenteritis or more severe cases, such as septicemia [2]. *Vibrio parahaemolyticus* (*V. parahaemolyticus*) is a Gram-negative, halophilic marine bacterium. It naturally occurs in seafood and is a principal etiological agent, responsible for bacterial infections linked to seafood consumption [3,4]. Consequently, *V. parahaemolyticus* represents a public health concern, contributing to seafood-borne illnesses across multiple Asian countries [5,6]. This underscores the urgent need for effective strategies to prevent the contamination of marine products by *V. parahaemolyticus*.

To mitigate the incidence of *V. parahaemolyticus* infections and prolong the shelf life of seafood products, the application of natural antimicrobial agents, such as organic acids and plant extracts, has been widely adopted as a strategic approach [7,8]. Xi et al. [9] demonstrated that the phenolic compounds present in a green tea extract could impede the growth of bacteria responsible for spoilage and diminish the incidence of infections caused by *V. parahaemolyticus* in oysters stored under refrigeration. Similarly, Sivasothy et al. [10] reported that flavonoids and curcuminoids exhibit substantial inhibitory activity against *V. parahaemolyticus*. These studies have demonstrated the feasibility of flavonoids to inhibit the survival of *V. parahaemolyticus* in seafood.

Phloretin (PHL), a flavonoid characterized by its dihydrochalcone structure, is found in apples and a range of apple-derived products [11]. According to the GB2760-2024 standard [12] for food additive use in China, phloretin is approved as a natural flavoring agent. Additionally, the Flavor and Extract Manufacturers Association of America (FEMA) has classified PHL as Generally Recognized as Safe (GRAS). This designation confirms its safety and reliability, suggesting its applicability in food preservation to extend the shelf life of food. The whitening, antioxidant, anti-inflammatory, antitumor, and anti-cancer activities of PHL have been well-documented in the scientific literature [13,14,15,16,17]. Some previous reports have indicated that PHL has antibacterial activity against a number of foodborne pathogens, including *Escherichia coli*, *Staphylococcus aureus* (*S. aureus*), and *Listeria monocytogenes* [18,19,20]. The current literature on the antibacterial effect of PHL on *V. parahaemolyticus* and its potential mechanism is still limited. Nevertheless, this information is crucial for the potential application of PHL in food as a natural bacteriostatic agent.

Herein, this study sought to gain a deeper understanding of the antibacterial mechanisms of PHL against *V. parahaemolyticus* at the molecular level. The effect of PHL on the appearance of the cell membrane, microstructure, enzymes, genetic material, and intracellular protein content of *V. parahaemolyticus* were examined in the research. The production of reactive oxygen species (ROS), which can cause cellular oxidative stress, was quantified. Additionally, this study explored the practical significance of PHL in restricting *V. parahaemolyticus* in seafood.

## 2. Materials and Methods

### 2.1. Materials and Reagents

#### 2.1.1. Materials

The strains of *V. parahaemolyticus* ATCC17802 were stored in the National Engineering Center of Seafood at Dalian Polytechnic University. Prior to use, the strains were revived and activated by trypticase soy broth (TSB, Hope Biotechnology, Qingdao, China), supplemented with 3% NaCl and cultured at a constant temperature of 37 °C for 18 h to attain the logarithmic phase of bacterial growth. The phloretin was from Shanghai Aladdin Biochemical Technology Co., Ltd. (Shanghai, China). The seafood samples (sea bass, shrimp, and oysters) were sourced locally from the Qian He market in Dalian, Liaoning Province, China.

#### 2.1.2. Reagents

The thiosulfate citrate bile salts sucrose (TCBS) agar medium was purchased from Qingdao Hope Bio-Technology Co., Ltd. (Qingdao, China). N-Phenyl-1-naphthylamine (NPN) was obtained from Shanghai Aladdin Biochemical Technology Co., Ltd. (Shanghai, China). The iodonitrotetrazolium chloride (INT) was from Shanghai Macklin Biochemical Technology Co., Ltd. (Shanghai, China). The resazurin, 2′, 7′-dichlorofluorescin diacetate (DCFH-DA) and propidium iodide (PI) solution (1 mg/mL) were from Beijing Solarbio Science & Technology Co., Ltd. (Beijing, China). The o-Nitrophenyl β-D-galactopyranoside (ONPG) was from Shanghai Yuanye Bio-Technology Co., Ltd. (Shanghai, China).

### 2.2. Phloretin Treatment

To ascertain the antibacterial efficacy of phloretin, the micro-well dilution method, as outlined by Tian et al. [21], was utilized to determine the minimal inhibitory concentration (MIC: 125 μg/mL). Briefly, the original PHL concentration was 500 μg/mL. PHL was dissolved in a sterile TSB medium containing 3% NaCl, and 100 μL of 2-fold serial dilution was added to each well of a 96-well microtiter plate, inoculated with 100 μL of activated *V. parahaemolyticus* (10^6^ CFU/mL), and incubated at 37 °C for 24 h. The growth of the bacterial cells was measured at OD_600nm_, using a multifunctional microplate detector (Tecan, Männedorf, Switzerland).

### 2.3. Time-Kill Analysis of PHL Against V. parahaemolyticus

The time-kill curve analysis of PHL against *V. parahaemolyticus* was tested using the plate colony counting method [22]. The incubation of the bacterial suspension at 37 °C with different PHL concentrations of 0 (control), 125, and 250 μg/mL for 0, 5, 30, and 60 min was conducted. The resulting concentration of the bacterial suspension was determined to be 10⁶ CFU/mL, using a phosphate buffer as the washing solution (PBS, 0.01 M, pH 7.2–7.4). Subsequently, a 10-fold serial dilution was performed. The diluent was then deposited onto a nutrient agar plate and incubated at 37 °C for 24 h. This allowed for the residual bacterial amount to be detected through the drop plate method.

### 2.4. Antibacterial Mechanism of PHL Against V. parahaemolyticus

#### 2.4.1. Effect of PHL on the Morphology of *V. parahaemolyticus*

The appearance and cell integrity of both the control and PHL-treated *V. parahaemolyticus* were observed with clarity, using a scanning electron microscope (SEM, JEOL, Tokyo, Japan) and a transmission electron microscope (TEM, JEOL, Tokyo, Japan).

The SEM sample processing steps were as follows: Different bacterial samples were immersed in a 2.5% glutaraldehyde solution (in 0.1 M phosphate buffer, pH 7.0) for approximately 12 h at 4 °C, for immobilization. Then, sequentially add 30%, 50%, 70%, 90%, and 100% ethanol for gradient elution. Each concentration was allowed to act for 10 min [23]. The dehydrated specimens were affixed to a brass sample stage treated with gold via conductive adhesives and, then, images were taken at an electron velocity of 10 kV to obtain the final image.

The TEM sample processing steps were as follows: The samples were subjected to centrifugation at 5000 rpm for 5 min, then washed three times with PBS solution. A copper mesh was immersed in the bacterial suspension and subsequently dried at room temperature. The bacteria present on the copper mesh were stained with phosphotungstic acid, thus facilitating observation under the TEM.

#### 2.4.2. Detection of Cell Damage

The use of PI staining facilitates the observation of cell damage [2]. The cells were harvested by centrifugation at 4 °C, washed, and resuspended in PBS. PHL stock solutions were added to the bacterial suspensions to reach concentrations of 0 (control) and 125 μg/mL, respectively, and treated for 1 h at 37 °C. The treated cells were washed, resuspended in PBS, stained with 50 μg/mL PI, and reacted for 5 min at 25 °C, in the dark. At the end of the reaction, 5 μL of untreated or PHL-treated bacterial suspension was dropped onto a microscope slide for visualization by the naked eye using a fluorescence microscope (Revole, Echo, San Diego, CA, USA). The stained bacterial cells were then analyzed (Ex 535 nm; Em 595 nm).

#### 2.4.3. Effect of PHL Treatment on the Cell Constituents of *V. parahaemolyticus*

##### The Leakage of Intracellular Nucleic Acid and Protein

The release of nucleic acid and protein into the cell suspension was measured by ultraviolet spectrophotometry and the Bradford method [22]. A cell suspension in PBS was mixed with 125 μg/mL PHL in an equal volume and inoculated at 37 °C for 60 min. Following this, the mixture was centrifuged and filtered through a 0.22 μm microporous membrane. The released nucleic acid content was quantified using a UV spectrophotometer (Metash, Shanghai, China), at a wavelength of 260 nm, while the released protein content was determined with a multifunctional microplate detector (Tecan, Männedorf, Switzerland), at a wavelength of 595 nm.

##### Measurement of the Outer Membrane (OM) and Inner Membrane (IM) Permeability

The *V. parahaemolyticus* in the logarithmic phase was collected following centrifugation at 5000 rpm for 5 min at 4 °C, washed, and resuspended in PBS buffer.

For the OM permeability: The outer membrane permeability of bacteria was monitored with the aid of NPN. The bacteria were mixed with NPN (150 μM) and the mixture was exposed to 0 (control) and 125 μg/mL PHL concentrations. The fluorescence intensity was determined using a fluorescent enzyme labeling instrument (Tecan, Männedorf, Switzerland) (Ex 360 nm; Em 465 nm).

For the IM permeability: The bacterial suspension, ONPG (1.5 mM), and 0 (control) or 125 μg/mL PHL concentrations were transferred to 96-well plates. The changes in the β-galactosidase activity were recorded using a spectrophotometer set to measure absorbance at a wavelength of 405 nm, allowing for a comparison of the results.

##### Measurement of Alkaline Phosphatase (AKP)

The content of AKP in the different groups was detected using an AKP assay kit (Jiancheng Bioengineering Institute, Nanjing, China) and the absorbance at 520 nm was measured using a multifunctional microplate detector (Tecan, Männedorf, Switzerland).

##### Measurement of ATP Concentration and the Leakage of Adenosine Triphosphatase (*ATPase*)

*V. parahaemolyticus* was treated with 125 μg/mL PHL and samples without PHL treatment were utilized as the control. The bacterial suspension was centrifuged at 10,000 rpm for 5 min, the supernatant was discarded, and the precipitate was collected and washed three times and resuspended in 0.9% saline solution [24]. Ultimately, the bacterial cells were disrupted using an ultrasonic cell crusher (Nanjing Emanuel Instrument Co., Ltd., Nanjing, China) and the ATP concentration was quantified, in accordance with the instructions provided with the ATP content determination kit (Jiancheng Bioengineering Institute, Nanjing, China).

To detect the *ATPase* activity, an *ATPase* assay kit (Jiancheng Bioengineering Institute, Nanjing, China) was utilized to ascertain the absorbance at 636 nm, via a multifunctional microplate detector (Tecan, Männedorf, Switzerland).

#### 2.4.4. Effect of PHL Treatment on the Vitality of *V. parahaemolyticus*

##### Metabolism

The metabolic levels were quantified using the fluorescent dye, resazurin. A solution of resazurin at a concentration of 100 μg/mL was added to the bacterial suspension, with or without prior treatment with PHL. Following a 1 h and 2 h incubation period in the dark, the samples were centrifuged at 10,000 rpm for 5 min. The resulting supernatants were then assayed using a fluorescence enzyme marker (Tecan, Männedorf, Switzerland) (Ex 535 nm; Em 595 nm).

##### Respiratory Chain Dehydrogenase

The respiratory chain dehydrogenase activity was quantified after PHL treatment. Briefly, bacterial suspensions were treated with 125 μg/mL PHL and incubated at 37 °C and 180 rpm/min for 1 and 2 h. The bacteria were centrifuged to collect the precipitate, which was mixed with INT at 37 °C for 30 min and rinsed with PBS. The respiratory chain dehydrogenase activity of *V. parahaemolyticus* was characterized by recording the absorbance at 630 nm, using a multifunctional microplate detector (Tecan, Männedorf, Switzerland).

#### 2.4.5. Measurement of Intracellular ROS After PHL Treatment

The ROS were identified through the use of a DCFH-DA fluorescent probe. The collected *V. parahaemolyticus* was resuspended in DCFH-DA solution (10 μmol/L) and incubated at 37 °C for 30 min, in the absence of light to permit the DCFH-DA to infiltrate the bacteria. After incubation, the mixture was washed to remove the residual free DCFH-DA and then treated with 125 μg/mL PHL. The sample without PHL treatment was used as a control. The fluorescence intensity was documented, employing the fluorescent enzyme labeling instrument (Tecan, Männedorf, Switzerland) (Ex 485 nm; Em 535 nm) [25].

#### 2.4.6. Effect of PHL on Intracellular Content of *V. parahaemolyticus*

##### Measurement of Protein Content

*V. parahaemolyticus* was washed and centrifuged with or without PHL treatment to collect precipitated bacteria, which were resuspended in 1 mL of sterile PBS. Subsequently, the protein in *V. parahaemolyticus*, treated with 125 μg/mL PHL for 1 h, was extracted by using a Gram-negative bacterial protein extraction kit (BestBio, Shanghai, China), according to the kit instructions. The protein content was ultimately determined through the use of a bicinchoninic acid (BCA) test kit (Beyotime, Shanghai, China) [26]. The measurement of the protein content from *V. parahaemolyticus* was conducted via sodium dodecyl sulfate–polyacrylamide gel electrophoresis (SDS-PAGE). The collected samples were combined with the sample buffer at a ratio of 4:1 and boiled for 10 min. Finally, the samples were centrifuged at 8000 rpm for 5 min, after which the supernatant was collected for the SDS-PAGE analysis [27].

##### Measurement of DNA Content

The bacterial genomic DNA in *V. parahaemolyticus* was extracted using the Solarbio Bacterial Genomic DNA Extraction Kit (Beijing, China), according to the instructions provided with the kit. Agarose gels were prepared by dissolving agarose (1%) in TAE buffer for subsequent gel electrophoresis. The integrity and purity of DNA binding can be directly observed through the use of a chemiluminescence imaging analysis system (Bio-Rad, Shanghai, China).

### 2.5. Application of PHL in Seafood Spiked with V. parahaemolyticus

After overnight growth, *V. parahaemolyticus* cells were harvested, washed, and resuspended in a solution containing 900 μL of TSB broth, containing 3% NaCl, to approximately 10^6^ CFU/mL. Manually separated seafood (comprising sea bass, shrimp, and oyster meat) was added to TSB broth containing 3% NaCl, then thoroughly mixed using a sterile homogenizer (Scientz, Ningbo, China) for 2 min. A 100 μL solution was absorbed and combined with the bacterial suspension. Subsequently, 125 μg/mL PHL was added to the homogenized samples, while the untreated samples served as the control. The homogenized samples were incubated at 37 °C for 1 h and diluted 10-fold in a gradient of TSB broth containing 3% NaCl, and 100 μL of the diluted solution was spread onto TCBS agar plates, which were incubated at 37 °C for 24 h. The *V. parahaemolyticus* colonies were counted and the counts were converted into logarithmic numbers, namely CFU/g [2].

### 2.6. Statistical Analysis

The data for each experiment were obtained by repeating the procedure at least three times and were expressed as the mean ± standard deviation (SD). A one-way analysis of variance (ANOVA) was performed on the results, using the IBM SPSS Statistics 26 software to determine the level of significance, with *p* < 0.05 considered significant.

## 3. Results and Discussion

### 3.1. Antibacterial Activity of PHL

The bactericidal activity of PHL was evaluated by using *V. parahaemolyticus* ATCC17802 as an indicator bacterium in the *V. parahaemolyticus* assay. The results showed that PHL had strong inhibitory activity against *V. parahaemolyticus*, with an MIC value of 125 μg/mL. Moreover, a time-kill assay was performed to further determine the bactericidal properties of PHL against *V. parahaemolyticus*. As shown in Figure 1A, the viability of *V. parahaemolyticus* was markedly reduced when compared to the control, exhibiting rapid inactivation, with a reduction in the viable count of up to 3.3 and 3.8 logs within 5 min. The results demonstrated the absence of viable cells in the treated area of the solid plate following treatment with PHL at concentrations of 125 μg/mL and 250 μg/mL for 15 min. This indicated that PHL exerted a robust and rapid bactericidal impact against *V. parahaemolyticus*. These findings highlight the potential application of PHL as a natural food preservative, due to its rapid and sustained bactericidal efficacy against *V. parahaemolyticus*.

### 3.2. Morphological Changes to V. parahaemolyticus

The morphological alteration of *V. parahaemolyticus* was documented through the use of SEM. As illustrated in Figure 1B, the untreated *V. parahaemolyticus* exhibited a bacilliform form, characterized by a complete surface and distinct outline. In opposition to the control group, *V. parahaemolyticus* treated with 125 μg/mL of PHL exhibited pronounced local shrinkage, with leakage of the intracellular contents, and evident cellular deformation. These results demonstrate that PHL had a disruptive effect on the cell wall and cell membrane of *V. parahaemolyticus*, leading to altered cell morphology and the release of the intracellular contents. This evidence substantiates the conclusion that the cell wall and cell membrane are disrupted. The observed phenomenon exhibited similarities to that reported by Diao et al. [28] in regard to monolauroyl-galactosyl glycerol against *Bacillus cereus*. To gain further insight into the intracellular alterations induced by *V. parahaemolyticus*, TEM was employed, with the results aligning with those observed through SEM. Figure 1C shows that the untreated cells retained their characteristic cellular architecture, including an intact cell wall and smooth membrane within the cytoplasm. In contrast, the PHL-treated bacteria are incomplete and broken.

The TEM and SEM analyses revealed that the PHL treatment resulted in destructive effects on the cell membranes of *V. parahaemolyticus*, manifesting in alterations to the cellular structure and disruption to the integrity of the cell walls and cell membranes, thereby leading to the leakage of the intracellular contents, which can be postulated to represent the antimicrobial mechanism of action of PHL against *V. parahaemolyticus*.

### 3.3. PHL Caused the Cell Damage to V. parahaemolyticus

The number of damaged or dead cells was quantified through fluorescent staining, with red dye indicating damaged cells (Figure 2A). There were very few cells with red fluorescence in the control group. In contrast, the number of cells with red fluorescence in the PHL-treated group was much higher. Meanwhile, the maximum fluorescence intensity of the PI was observed in the control group at 18,148.33 a.u., whereas the treated samples demonstrated a maximum fluorescence intensity of 27,504.67 a.u. (Figure 2B). These findings suggest that PHL may have enhanced the number of damaged or dead cells in *V. parahaemolyticus*.

### 3.4. PHL Induced the Leakage of Intracellular Nucleic Acid and Protein

The disruption of the cell membrane, which serves as the external barrier, can result in the leakage of a multitude of crucial biomolecules (protein, nucleic acid, etc.) from the bacteria. This consequently impacts the normal anabolic function of cells, thus prompting the measurement of intracellular substance leakage. The results, as illustrated in Figure 3A, demonstrate that the values measured at 260 nm exhibited a notable increase following PHL treatment. This suggests that nucleic acid may have escaped from the compromised membranes, potentially impeding gene expression and various enzyme-related metabolic processes. In parallel, the leakage of protein was determined. As evidenced by Figure 3B, PHL resulted in protein leakage. The observed outcome indicates that PHL has the potential to induce the formation of transmembrane channels in the cell membrane, which could potentially lead to the release of small molecules from the inside the cell. The application of mannosylerythritol lipids, studied by Shu et al. [29] against *S. aureus*, also resulted in the leakage of intracellular components through a similar mechanism of action.

### 3.5. PHL Increased the Release of AKP by V. parahaemolyticus

AKP is present in the space between the bacterial cell wall and the cell membrane. In general, the enzyme is not released into the extracellular space. However, when bacteria are subjected to unfavorable conditions, such as the action of bacteriostatic agents, the permeability of the cell wall may increase, resulting in the release of AKP. In terms of the extracellular AKP activity (Figure 3C), the extracellular AKP activity of the PHL-treated bacteria was 7.17 U/L, significantly increasing the extracellular level of the AKP enzyme compared to the control group (which was 3.65 U/L) after PHL treatment. This indicates that PHL treatment resulted in the destruction of the bacteria, which in turn caused AKP to leak into the extracellular space and the extracellular level of the AKP enzyme to increase.

### 3.6. The Impact of PHL on OM and IM Permeability of V. parahaemolyticus

NPN is a fluorescent probe that exhibits hydrophobic properties. When the outer membrane of a bacterial cell is impaired or its structural integrity is modified, NPN is able to traverse the outer membrane and enter the hydrophobic environment of the cell, leading to an enhancement in the fluorescence intensity. It has been described in the literature that an increase in the concentration of bacteriostatic substances or the reaction time, when the cell membrane of bacterial cells is severely damaged, may also cause NPN to escape from the hydrophobic environment, which leads to a decrease in the fluorescence absorption value [30]. As illustrated in Figure 4A, the fluorescence intensity of the extracellular membrane of *V. parahaemolyticus* exhibited a notable elevation subsequent to PHL treatment, which indicates that the extracellular membrane of *V. parahaemolyticus* was in a state of instability and weakness.

β-Galactosidase is a hydrolase found in the inner membrane or inside the cell that catalyzes the hydrolysis of lactose into galactose and glucose to meet the demands of bacterial growth [31]. ONPG is an analog of lactose. The hydrolysis of β-galactosidase into O-nitrophenol is accompanied by an absorptive peak at 405 nm. Therefore, a change in absorbance at 405 nm may reflect changes in the β-galactosidase activity. As seen in Figure 4B, the OD_405nm_ value of PHL-treated bacteria increased significantly compared to the control group. This result indicates that the activity of β-galactosidase decreased and the bacterial metabolic capacity was impaired after PHL treatment, which ultimately influenced the growth rate of the bacteria.

### 3.7. PHL Influenced the ATP Content and Na^+^K^+^-ATPase Activity of V. parahaemolyticus

The differences in the ATP content and Na^+^K^+^-*ATPase* activity are shown in Figure 5. The activity of Na^+^K^+^-*ATPase* declined by 43.25% in *V. parahaemolyticus* treated with PHL compared to the control group (Figure 5B). These findings indicate that PHL is capable of impeding the activity of Na^+^K^+^-*ATPase*, which plays a pivotal role in bacterial growth. Na^+^K^+^-*ATPase* is a globular protein, whose catalytic activity relies on the native configuration of its active sites and the conformation of the surrounding proteins [32]. PHL has the potential to induce alterations in the protein structure, which may result in a decline in enzyme activity. Moreover, the decline in Na^+^K^+^-*ATPase* activity resulted in a corresponding decline in ATP content within the bacterial cells, as evidenced in Figure 5A. A similar outcome was observed in a study conducted by Han et al. [33], where the activity of Na^+^K^+^-*ATPase* was found to decline significantly following limonene treatment.

### 3.8. PHL Modulated the Cell Metabolism of V. parahaemolyticus

Figure 6 illustrates the metabolic alterations in *V. parahaemolyticus* following PHL treatment. Once it has gained access to the interior of the bacterium, resazurin, a weak fluorescence indicator compound, is degraded by a range of oxidoreductases into a molecule with significantly greater fluorescence, resorufin. As seen in Figure 6A, the reduction in fluorescence intensity over the course of 1 and 2 h after PHL treatment suggests a gradual decrease diminution in the metabolic activity of *V. parahaemolyticus*. However, no bacterial growth was observed following a 15 min incubation period during the time-kill analysis. The bacteria were in a sublethal state, as evidenced by their inability to be cultured on a solid medium, while still maintaining metabolic activity. This phenomenon has been previously observed and documented in the literature [34].

In order to gain further insight into the impact of PHL treatment on metabolic activity, the respiratory chain dehydrogenase activities within the treated samples were determined by INT. Respiratory chain dehydrogenase reduces INT to a dark red water-insoluble substance and the activity of dehydrogenase is determined by colorimetry. As demonstrated in Figure 6B, the addition of PHL resulted in a notable reduction in respiratory chain dehydrogenase activity, indicating an increase in energy restriction, consistent with the deceased metabolic activity determined by resazurin.

### 3.9. PHL Elevated the Intracellular ROS Level of V. parahaemolyticus

The results demonstrate that intracellular ROS production is significantly enhanced following PHL exposure, as illustrated in Figure 7. It is postulated that elevated ROS levels may impede the functionality of specific organelles, potentially leading to structural damage within the *V. parahaemolyticus* cells.

### 3.10. The Effect of PHL Treatment on the Intracellular Protein in V. parahaemolyticus

In bacterial cells, protein macromolecules serve a variety of functions, including that of catalysts and structural support agents, thereby maintaining cell viability and structural integrity [35]. The release of intracellular protein and changes in protein content can further confirm the integrity of the cellular structure [36].

To ascertain whether any differences in protein composition existed between the treated and untreated samples, a quantitative analysis was conducted. The results of the analysis are illustrated in Figure 8A. Following treatment with PHL, the protein concentration of the bacterial sample was found to be 510.0 μg/mL. This represents a decrease of 14.02% in the protein content when compared with the control group, which demonstrated a protein concentration of 592.4 μg/mL. This outcome demonstrated a considerable reduction in soluble proteins in the bacterial cells following treatment with PHL. The underlying mechanism behind the disruption may be attributed to PHL’s ability to alter the cellular structure of *V. parahaemolyticus*, leading to the release of intracellular proteins into the surrounding environment [37]. The finding is inconsistent with the results reported by Chen et al. [38], which demonstrated that the protein bands of *V. parahaemolyticus* exhibited no discernible alteration after treatment with curcumin. Given the diversity of their chemical structures, varying degrees of antimicrobial sensitivity and the numerous mechanisms of antimicrobial activity, different phenolic compounds are likely to exhibit disparate modes of antibacterial action.

### 3.11. PHL Decreased the Genomic DNA in V. parahaemolyticus

DNA is the primary genetic material that regulates fundamental life processes, encompassing aspects such as bacterial growth and heredity. In the absence of external factors, the detection of DNA outside of bacterial cells is not possible unless the bacterial cell has suffered damage by an external force. According to Figure 8B, the DNA concentration in PHL-treated bacteria was markedly diminished in comparison to the control. It is evident from the data that the DNA concentration in the bacteria, namely *V. parahaemolyticus*, decreased from 45.9 to 32.1 μg/mL over the course of the experiment. The observed decrease in DNA concentration may be attributed to the fact that the PHL treatment resulted in structural damage to the bacteria, ultimately leading to a significant loss of DNA.

Agarose gel electrophoresis further validated the above results. Figure 8C shows that the brightness in the experimental group was markedly diminished in comparison to the control group, indicating a notable reduction in the DNA content in the bacteria following PHL treatment. The administration of PHL results in the acceleration of the exudation of DNA from compromised cell membranes, or it impedes the synthesis of DNA in the organism, *V. parahaemolyticus*. Consequently, the DNA content in the PHL-treated group was found to be lower than that observed in the control group. These results align with the observations by Hu et al. [39], which indicated that juglone can induce the reduction and leakage of DNA, ultimately leading to the degradation or disappearance of bacterial DNA.

### 3.12. PHL Effectively Inactivated V. parahaemolyticus in Seafood

The potential applications of PHL as a natural antimicrobial in seafood were investigated. One of the principal causes of seafood-associated, foodborne diseases is the presence of *V. parahaemolyticus* in seafood, such as shellfish and shrimp [38,40]. The objective of the present study was to evaluate the antibacterial activity of PHL against *V. parahaemolyticus* in seafood meats. The control treatment exhibited no statistically significant differences in the *V. parahaemolyticus* counts in sea bass fillet, shrimp, and oysters. The application of PHL resulted in a notable reduction in the bacterial cell count in oysters, as evidenced by a 2.5-log reduction in CFU/g (Figure 9A). A comparable decline in the bacterial load was observed in sea bass fillet and shrimp, with PHL treatment leading to a 2.2- and 2.0-log reduction in bacteria (Figure 9B,C), respectively. These outcomes suggest the efficacy of PHL in controlling bacterial growth in various seafood species.

PHL can swiftly and effectively impart robust antimicrobial properties onto seafood. Nevertheless, the bactericidal effect of PHL applied to each seafood species (shrimp > sea bass fillet > oysters) was different. This situation can be attributed to the inherent variability of seafood, such as ionic strength, pH, water activity, and the composition of proteins and fats, which vary from one another [41]. The complexity of the food matrix may have a considerable impact on the distribution of PHL to bacteria, which in turn affects its bactericidal activity [42]. The reduction in *V. parahaemolyticus* in the tested seafood after treatment with 250 μg/mL of PHL reached 2.0 to 2.5 log_10_ CFU/g. This is in accordance with US FDA guidelines, which require that the total number of *V. parahaemolyticus* in seafood products should be reduced by a level of 2 to 3 log during post-harvest processing [2].

## 4. Conclusions

In summary, we demonstrated here that PHL impacted the cell morphology, compromised the cell integrity of *V. parahaemolyticus*, and decreased the protein, ATP, and DNA contents in *V. parahaemolyticus*. Moreover, PHL inhibited the activity of several key enzymes and metabolism in *V. parahaemolyticus* and caused an elevation in the intracellular ROS concentration. Finally, PHL successfully reduced the number of *V. parahaemolyticus* cells in seafood (sea bass, shrimp, and oysters). These findings suggest that PHL could potentially be developed as a natural preservative to reduce contamination with *V. parahaemolyticus* in seafood, thereby broadening the range of application of PHL in the food industry. Further research is necessary to determine the appropriate concentration and its impact on the organoleptic properties of foods prior to its application in real scenarios.

## Figures and Tables

**Figure 1 foods-13-03537-f001:**
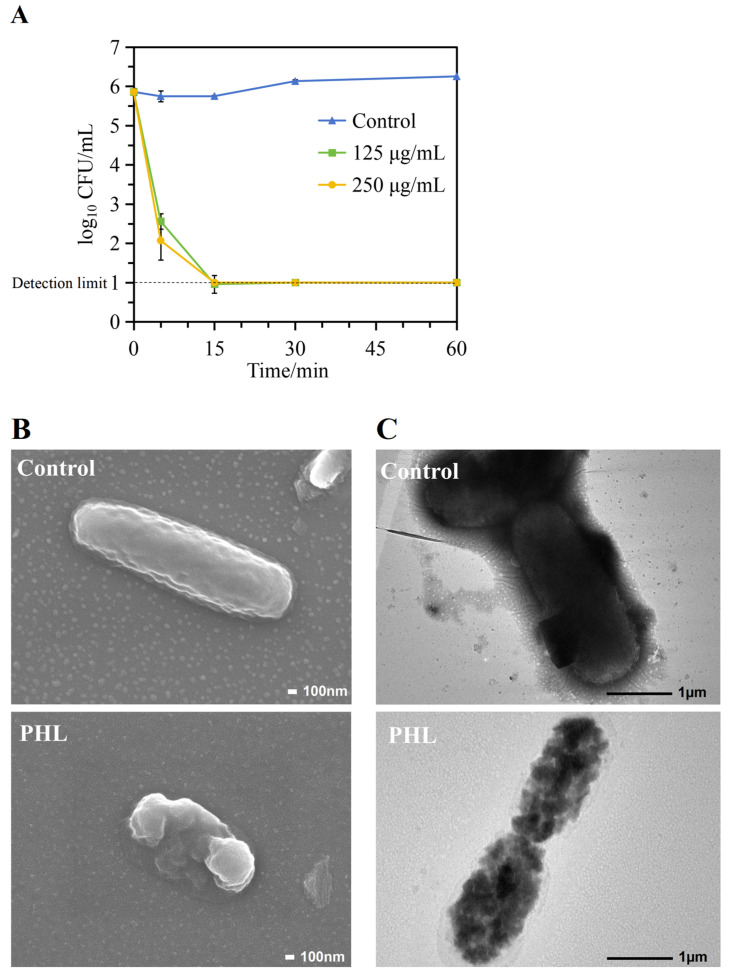
(**A**) Time-kill curve of PHL at different concentrations against *V. parahaemolyticus* ATCC18702. (**B**) SEM pictures of control and PHL-treated *V. parahaemolyticus* ATCC18702 at 30,000× magnification. (**C**) TEM pictures of control and PHL-treated *V. parahaemolyticus* ATCC18702 at 10,000× magnification. Data are the mean ± standard deviation of three experiments.

**Figure 2 foods-13-03537-f002:**
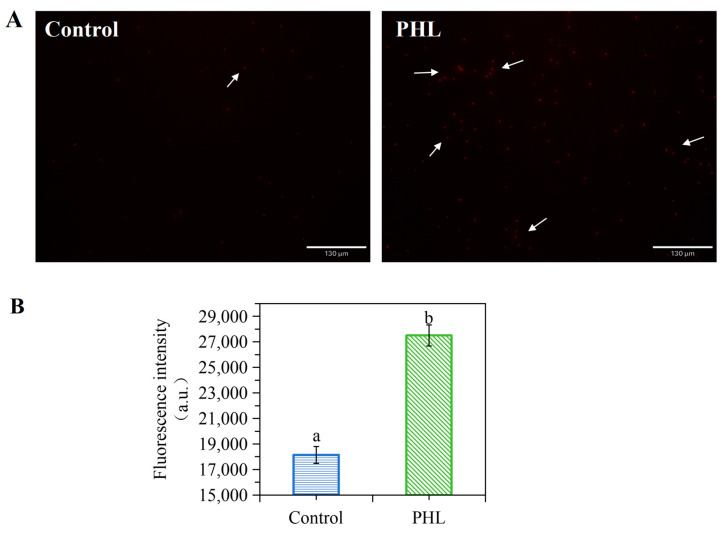
The effect of PHL on the cell damage to *V. parahaemolyticus*. (**A**) Fluorescence microscope images of *V. parahaemolyticus* treated with PHL. The area of red fluorescence is indicated by the arrow direction (→). (**B**) Fluorescence intensity of PI in *V. parahaemolyticus* cells of control and PHL-treated group. The a and b represent the significant difference between the two samples.

**Figure 3 foods-13-03537-f003:**
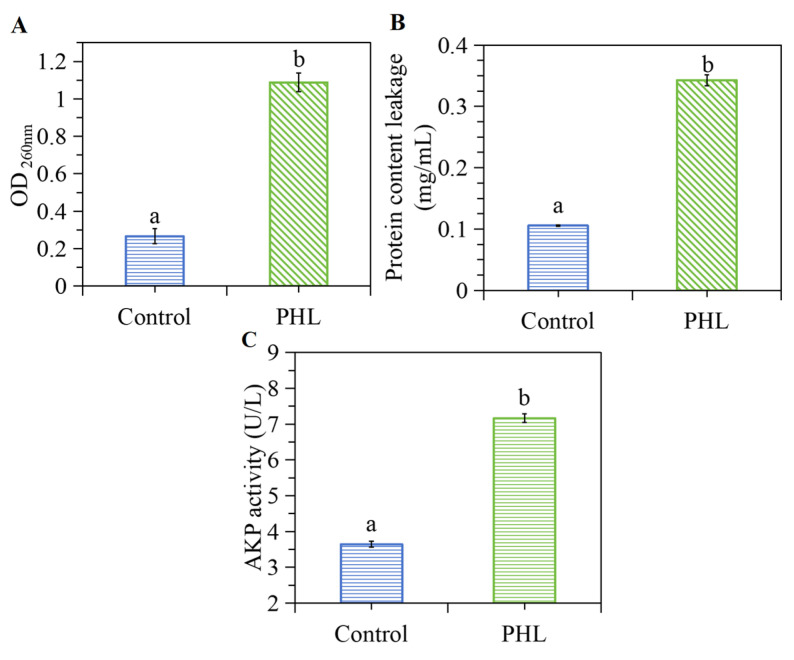
The effect of PHL on extracellular nucleic acid (**A**), protein (**B**), and AKP activity (**C**) of *V. parahaemolyticus* ATCC17802. The a and b represent the significant difference between the two samples.

**Figure 4 foods-13-03537-f004:**
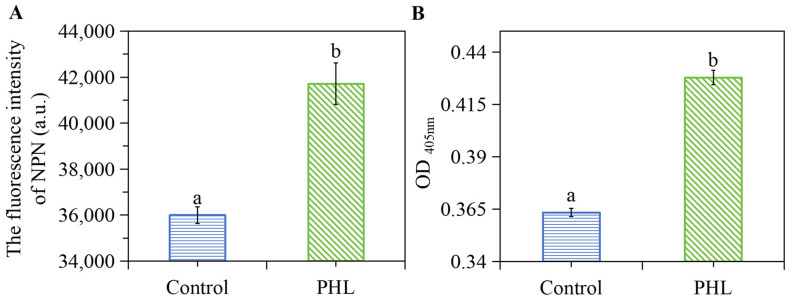
The effect of PHL on NPN uptake (**A**) and β-galactosidase (**B**) of *V. parahaemolyticus* ATCC17802. The absorbance change in terms of OD_405nm_ indicates the activity of β-galactosidase. The a and b in (**A**,**B**) represent the significant difference between the two samples.

**Figure 5 foods-13-03537-f005:**
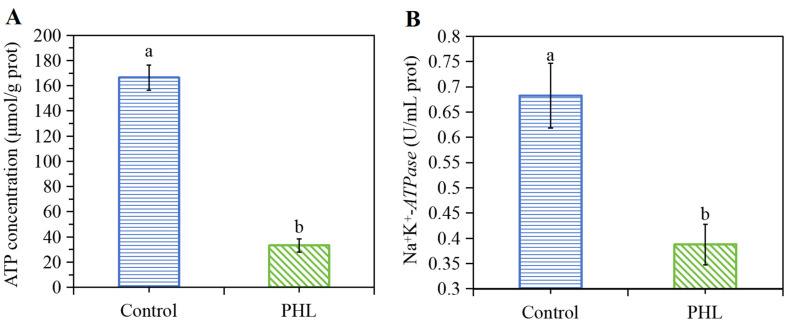
The change in (**A**) ATP concentration and (**B**) Na^+^K^+^-*ATPase* activity of bacteria treated with PHL. The a and b in (**A**,**B**) represent the significant difference between the two samples.

**Figure 6 foods-13-03537-f006:**
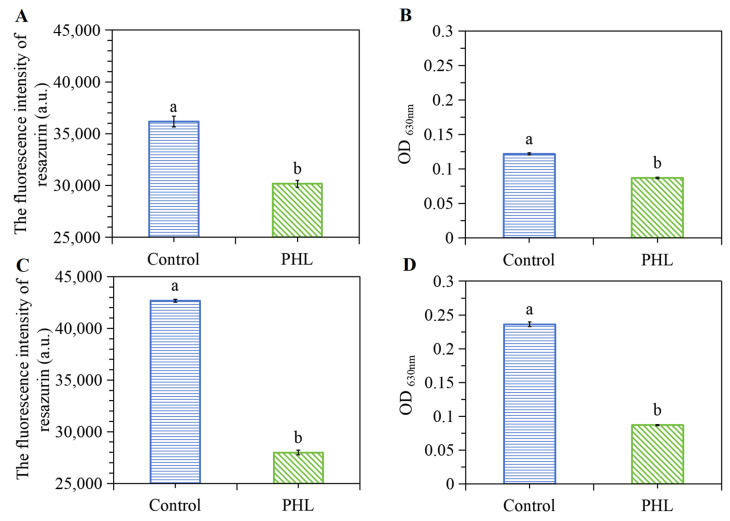
The effect of PHL on resazurin uptake and respiratory chain dehydrogenase of *V. parahaemolyticus* ATCC17802 after 1 h (**A**,**B**). The effect of PHL on resazurin uptake and respiratory chain dehydrogenase of *V. parahaemolyticus* ATCC17802 after 2 h (**C**,**D**). The absorbance change in terms of OD_630nm_ indicates the activity of respiratory chain dehydrogenase. The a and b represent the significant difference between the two samples.

**Figure 7 foods-13-03537-f007:**
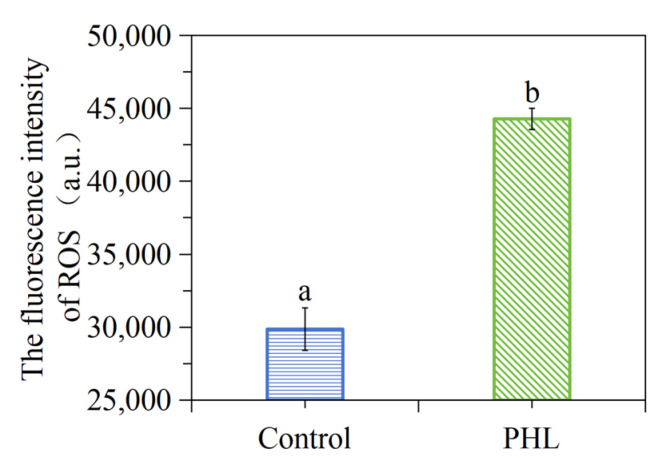
Detection of intracellular ROS after PHL treatment in *V. parahaemolyticus* ATCC17802. The a and b represent the significant difference between the two samples.

**Figure 8 foods-13-03537-f008:**
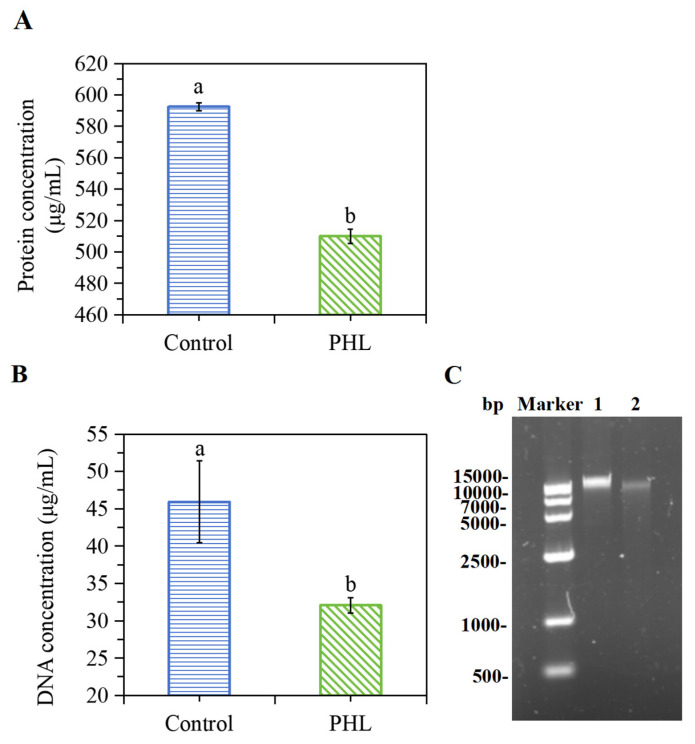
The intracellular protein concentration (**A**) of the bacterial protein in *V. parahaemolyticus* ATCC17802 following PHL treatment. The intracellular DNA concentration (**B**) and agarose gel electrophoresis pattern of genomic DNA (**C**) in *V. parahaemolyticus* ATCC17802. Lane 1: control; Lane 2: PHL treatment. The a and b represent the significant difference between the two samples.

**Figure 9 foods-13-03537-f009:**
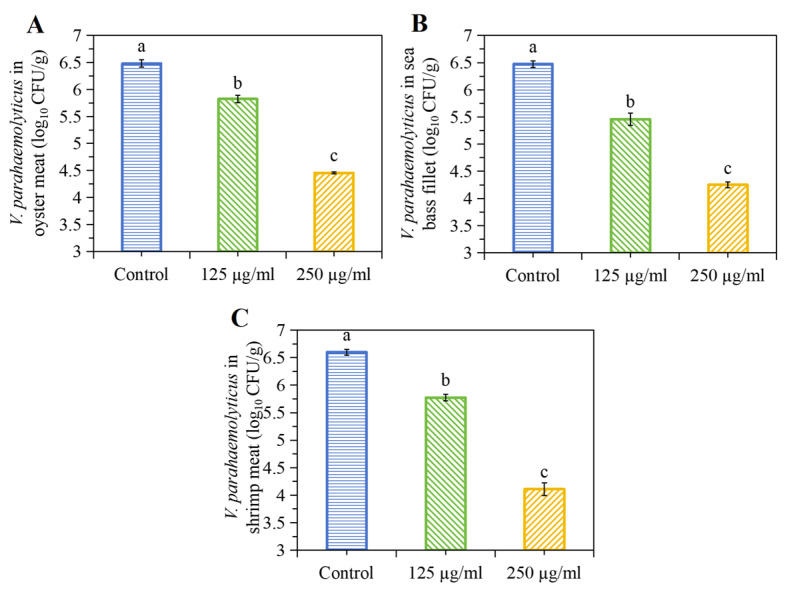
The effects of PHL on *V. parahaemolyticus* inactivation in seafood: (**A**) oyster meat, (**B**) sea bass fillet, (**C**) shrimp meat. Different letters (a, b, c) on the columns indicate the significant difference between each other.

## Data Availability

The original contributions presented in this study are included in the article. Further inquiries can be directed to the corresponding author.

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
