# Peer review of "Antibacterial Activity of Phloretin Against Vibrio parahaemolyticus and Its Application in Seafood"

_foods, 2024, doi:10.3390/foods13223537_

Round 1

Reviewer 1 Report

Comments and Suggestions for Authors

Dear authors

This manuscript examines the antibacterial activity of phloretin against V parahemolyticus and its application in seafoods. The paper is largely well written but there are a number of areas for improvement. The number of Figures at 12 is quite a lot, consider reducing the number?

Should the title be Antibacterial not Antimicrobial as you have tested against fungi, yeast or viruses?

Abstract

Line 9 its impact on foodborne pathogen (s)

Line 18 When "bening applyied" in sea bass, change to being applied

Line 222 was evaluated by using PHL as an indicator bacterium in the V. parahemolyticus assay?

Line 245 How do you know protoplasts were released?

Figure 2 the TEM image of control cells could be better?

Line 278 PHL induced the he leakage of intracellular nuclear acid and protein?? 

Line 280 leakage of a multitude of crucial biomolecules (proteins, nucleic acids, etc.) within the bacteria, change within to from

Line 315 does the legend fully explain the graph in Figure 5. The B-galactosidase activity is poorly explained.

Line 329 Legend needs more detail

Line 362 Consistent with the findings resazurin. This need to be rephrased

Line 365 The a and b in Fig. 8A and B meaned significant difference. Needs to be revised.

Line 403 The SDS-PAGE data doesnt add to the work as it is difficult to see changes in banding pattern. 

Line 433  "diseases is" delete extra space 

Line 458 The study sought to examine the antibacterial activity and antimicrobial, how did it examine both?

Comments on the Quality of English Language

English is good overall

Author Response

Response to Reviewer 1 Comments:

This manuscript examines the antibacterial activity of phloretin against V parahemolyticus and its application in seafood. The paper is largely well written but there are a number of areas for improvement. The number of Figures at 12 is quite a lot, consider reducing the number?

---- Thanks for your suggestion. We adjusted the figure layout and reduced the number of figures from 12 to 9.

Should the title be Antibacterial not Antimicrobial as you have tested against fungi, yeast or viruses?

---- Thank you for your comments. We agree with you and have revised the Antimicrobial to Antibacterial.

Abstract

Line 9 its impact on foodborne pathogen (s)

---- Thank you for your serious comments. We are very sorry for our careless mistake and it was rectified at its impact on foodborne pathogens.

Line 18 When "bening applyied" in sea bass, change to being applied

---- Thanks. We have changed “bening applied” to “being applied”.

Line 222 was evaluated by using PHL as an indicator bacterium in the V. parahemolyticus assay?

---- Thanks. We have corrected “using PHL as an indicator bacterium in the V. parahemolyticus assay” to “using V. parahaemolyticus ATCC 17802 as an indicator bacterium in the V. parahaemolyticus assay”

Line 245 How do you know protoplasts were released?

---- Thank you for your question. Based on your comment we have modified it to intracellular contents.

Figure 2 the TEM image of control cells could be better?

---- Thank you for your concern. During the experiments, we tried several times and the TEM image of control cell used in the manuscript was currently the clearest one we had. We enlarged the image and make it clearer in the revised manuscript. In future studies, we will strive to optimize experimental conditions to improve the TEM image quality.

Line 278 PHL induced the he leakage of intracellular nuclear acid and protein?? 

---- Thanks. We deleted “he”.

Line 280 leakage of a multitude of crucial biomolecules (proteins, nucleic acids, etc.) within the bacteria, change within to from

---- Thanks. We have revised it as suggested.

Line 315 does the legend fully explain the graph in Figure 5. The B-galactosidase activity is poorly explained.

---- Thanks. The original Figure 5 was adjusted to Figure 4. We have supplemented and refined the relevant information in the legend.

Line 329 Legend needs more detail

---- Thanks. We incorporated the figure into Figure 3 and added more details in the legend.

Line 362 Consistent with the findings resazurin. This need to be rephrased

---- Thanks. We have corrected it to “ indicating an increase in energy restriction, consistent with the deceased metabolic activity determined by resazurin.”.

Line 365 The a and b in Fig. 8A and B meaned significant difference. Needs to be revised.

---- Thanks. The original Figure 8 was adjusted to Figure 6. And the original figure and legend were modified.

Line 403 The SDS-PAGE data doesnt add to the work as it is difficult to see changes in banding pattern. 

---- Thanks. After considering your suggestion, we removed the SDS-PAGE result in the revised manuscript.

Line 433  "diseases is" delete extra space 

---- Thanks. We have deleted extra space.

Line 458 The study sought to examine the antibacterial activity and antimicrobial, how did it examine both?

---- Thanks. We misinterpreted the meaning of antimicrobial and have revised the Conclusions section in the revised manuscript.

Reviewer 2 Report

Comments and Suggestions for Authors

Comments reviewer

I found the article interesting, but it requires a few changes.

Title

        - parahemolyticus should be corrected to parahaemolyticus

        The formatting for author affiliations and contact details is lacking in the form

Abstract

        The abstract needs modifying as it is lacking in academic formulations. The writing should also be reviewed as there are grammatical errors: incorrect use of singular/plural forms, “bening applyied” should be corrected to “being applied.”

Introduction

-The introduction should be rewritten to ensure the flow of ideas as it seems to change from idea to idea without much connection.

 - Vibrio parahemolyticus should be corrected to Vibrio parahaemolyticus throughout.

Since the microorganism is the only bacterial strain to be researched in the manuscript, I believe it is necessary to be careful and use the name correctly.

Materials and methods

-All the details regarding materials used must be defined. For example the producers and country of origin of TSB,

Other chemical agents used in this study were of analytical reagent grade, what producer?

Seafoods are sourced from the nearby market, locally known as Qian He – reformulate perhaps as Seafood samples were sourced locally from the Qian He market in Dalian, China.

-In Line 165, should be mL

-In line 147 if you used a commercial kit, please fill in with the details

-Line 163 Please use italics when referencing microorganisms

-Same in 174, 184

Results

 - Please modify the concentrations tested to be expressed as the actual concentration and not as MIC 0, MIC 1 or MIC 2

 - Ensure that all figures contain clear error bars and indicate statistical significance directly in the figure legends

 - Figure 2 should include magnification levels clearly

 - Figure 3 must be improved as it is unclear and challenging to see what it represents to the reader. Higher resolution images with clear representation of magnification/scale bar

Conclusions

The conclusion could be reformulated to be more specific about the potential applications of PHL. Include a sentence on what is required to move from laboratory findings to practical food safety applications.

Author Response

Response to Reviewer 2 Comments:

I found the article interesting, but it requires a few changes.

Title

        - parahemolyticus should be corrected to parahaemolyticus

---- Thanks for your comment. We have corrected all parahemolyticus to parahaemolyticus in the manuscript.

        The formatting for author affiliations and contact details is lacking in the form

---- Thanks. We have added the author's affiliation in line 4 and 5.

Abstract

        The abstract needs modifying as it is lacking in academic formulations. The writing should also be reviewed as there are grammatical errors: incorrect use of singular/plural forms, “bening applyied” should be corrected to “being applied.”

---- Thanks for your comments. We have thoroughly corrected language issues and had the manuscript proof-read by a native English speaker.

Introduction

-The introduction should be rewritten to ensure the flow of ideas as it seems to change from idea to idea without much connection.

---- Thanks for pointing this out. We have revised the introduction part as suggested to make the ideas flow more smoothly.

 - Vibrio parahemolyticus should be corrected to Vibrio parahaemolyticus throughout.

Since the microorganism is the only bacterial strain to be researched in the manuscript, I believe it is necessary to be careful and use the name correctly.

 ---- Thank you very much for your comments, we have corrected the full text of parahemolyticus to parahaemolyticus. Thank you very much for your reminding. In the future research, I will be careful and use the name correctly.

Materials and methods 

-All the details regarding materials used must be defined. For example the producers and country of origin of TSB,

Other chemical agents used in this study were of analytical reagent grade, what producer?

---- Thanks. We have added TSB and other chemical reagent producers and countries to the manuscript.

Seafoods are sourced from the nearby market, locally known as Qian He – reformulate perhaps as Seafood samples were sourced locally from the Qian He market in Dalian, China.

---- Thanks. We have changed it to “Seafood samples (sea bass, shrimp, and oysters) were sourced locally from the Qian He market in Dalian, Liaoning Province, China.”

-In Line 165, should be mL

---- Thanks. We have corrected it.

-In line 147 if you used a commercial kit, please fill in with the details

---- Thanks. We have added kit producers and countries to the revised manuscript.

-Line 163 Please use italics when referencing microorganisms

-Same in 174, 184

 ---- Thanks. We have corrected them to italics.

Results

 - Please modify the concentrations tested to be expressed as the actual concentration and not as MIC 0, MIC 1 or MIC 2

---- Thanks. We have modified MIC 0, MIC 1 or MIC 2 to 0, 125 and 250 μg/mL.

 - Ensure that all figures contain clear error bars and indicate statistical significance directly in the figure legends

---- Thanks. We have made all error bars legible and indicated the statistical significance in the legends.

 - Figure 2 should include magnification levels clearly

---- Thanks. We have added the electron microscope image from Figure 2 to Figure 1, and improved the quality of the image so that the magnification and scale bar can be clearly seen.

 - Figure 3 must be improved as it is unclear and challenging to see what it represents to the reader. Higher resolution images with clear representation of magnification/scale bar

 ---- Thanks. We have modified the original Figure 3 and indicate the red fluoresence with arrows. In addition, the clear scale bar was provided on the image.

Conclusions

The conclusion could be reformulated to be more specific about the potential applications of PHL. Include a sentence on what is required to move from laboratory findings to practical food safety applications.

---- Thank you for your suggestion. We have revised it as follows: In summary, we demonstrated here that PHL impacted on the cell morphology, compromised cell integrity of V. parahaemolyticus, and decreased the protein, ATP, and DNA contents in V. parahaemolyticus. Moreover, PHL inhibited the activity of several key enzymes and metabolism in V. parahaemolyticus, and caused an elevation of the intracellular ROS concentration. Finally, PHL successfully reduced the number of V. parahaemolyticus cells in seafood (sea bass, shrimp, and oysters). These findings suggest that PHL could be potentially developed as a natural preservative to reduce contamination with V. parahaemolyticus in seafood, thereby broadening the range of application of PHL in food industry. Further research is necessitated to determine the appropriate concentration and its impact on organoleptic properties of foods prior to its application in real scenarios.

Reviewer 3 Report

Comments and Suggestions for Authors

Comment on the manuscript “Antimicrobial activity of phloretin against Vibrio parahemolyticus and its application in seafoods” by Siyang Chen, Wenxiu Zhu, Yiqun Zhan and Xiaodong Xia

The manuscript “Antimicrobial activity of phloretin against Vibrio parahemolyticus and its application in seafoods” by Siyang Chen and co-workers deals with an important topic. However, there are problems with the manuscript. The objective of this study was to examine the deeper understanding of the antibacterial mechanism of phloretin against V. parahemolyticus. The article's principal strength lies in the fact that the antimicrobial substance in question has not previously been subjected to such comprehensive study.

General comments:

The weakest part of the manuscript is the chapter on Materials and methods, which is written with many gaps. The work does not ensure reproducibility.

Specific comments:

Line 18: beeing instead of bening

Line 19: The colony counting method is not accurate enough to give the result to two decimal places.

Line 29: Gram-negative instead of gram-negative

Line 36: „to avoid or remove „ - How can the contamination of V. parahemolyticus be removed from a food? I suggest to delete „or removed”.

Line 41: Please specify, which tea extract it is.

Line 65: of cell instead of ofcell

Line 74: Please provide producer, country and product number.

Lines 76-77: „Other chemical agents used in 76 this study were of analytical reagent grade. „Please provide producer and country.

Line 77: Please specify the type of seafoods here.

Line 82: Add the concentration of PHL in the initial solution.

Lines 109-110: then washed three times with PBS solution. instead of after which they are washed on three occasions with a PBS solution.

Line 123: V. parahemolyticus (italic) instead of V. parahemolyticus

Line 127: Add the concentration of PHL.

Line 139: Specify what kind of fluorescent enzyme labeling instrument was used.

Line 147-149: Specify the producer and country of the AKP assay kit and microplate detector.

Line 152: Add the concentration of PHL.

Line 155: In what was it resuspended?

Line 162: Provide producer and country for microplate detector.

Line 163: V. parahemolyticus (italic) instead of V. parahemolyticus

Line 168: Specify what kind of fluorescence enzyme marker was used.

Line 171: Add the concentration of PHL.

Line 172: Please indicate the conditions under which the incubation period was 1 hour and the conditions under which it was 2 hours.

Line 173: Please specify what does INT present.

Line 174: V. parahemolyticus (italic) instead of V. parahemolyticus

Line 175: Specify the producer and country of the microplate detector.

Line 178: Please specify the producer and country and product number for the fluorescent probeV. parahemolyticus (italic) instead of V. parahemolyticus

Line 181: Add the concentration of PHL.

Line 183: Specify what kind of fluorescent enzyme labeling instrument was used.

Line 184: V. parahemolyticus (italic) instead of V. parahemolyticus

Line 188: the protein instead of The protein; add the concentration of PHL.

Line 189: Gram instead of gram; please specify the producer and country and product number of the protein extraction kit.

Lines 190-191: Please specify the producer and country and product number of bicinchoninic acid (BCA) test kit.

Line 204: V. parahemolyticus cells instead of V. parahemolyticuscells cells

Lines 204-208: Please indicate at what initial microbial count was the pathogen inoculated?

Line 208: Add the concentration of PHL.

Lines 211-212: was spread on TCBS agar plates, which were incubated  instead of  was applied to the surface. The TCBS solid plates were inoculated with a blotting rod, and the plates were incubated

Line 213: The V. parahemolyticus colonies were counted  instead of  The V. parahemolyticus cells colonies visible were counted

Line 222: “using PHL as an indicator bacterium” PHL is not a bacterium. Please reconsider.

Line 228: 3.8 instead of 3.79

Figure 1.: log10 CFU/ml instead of log CFU/ml

Line 256: which can be can be postulated  instead of  which can be can be postulated

Line 264: PHL caused the cell damage  instead of  The caused the cell damage

Lines 266-269: The following sentences (or parts of them) are may have been inserted due to a copy error, please correct. A significantly lower number of red blood cells were identified in individuals within the control group. In contrast, the proportion of red blood cells in the PHL-treated group was observed to be higher.

Figure 3: The fluorescence microscope images of V. parahemolyticus are not visible, and there is only an empty black square in parts A and B. These figures should be replaced.

Line 283: „all 260 absorbance values exhibited a notable increase” – this is not correct. The values measured at 260 nm exhibited a notable increase

Line 284: This suggests  instead of  The suggests

Line 312-313: This indicates instead of The indicates

Lines 353-354: The following is not clear. „the bacteria were not detected. Bacteria were certainly present, possibly not alive/not culturable. Please correct.

Line 359: In order to establish this, instead of To this end,

Line 362: restriction, consistent   instaed of  restriction. Consistent

Lines 383-384: In my opinion, quite poorly, the marked contrast between the control and PHL-treated groups mentioned in the text was observed. I don't know if it is possible to improve the quality of the gel image afterwards, but it is not conclusive.

Lines 387-388: “the protein bands in the SDS-PAGE transformed light to dark” Where could this be seen?

Line 389: The finding is inconsistent with the results reported by Chen et al. instead of The finding is at odds with that reported by Chen et al.

Line 432: were investigated instead of was investigated

Line 439: a 2.5-log reduction  instead of  a 2.48-log reduction, use one decimal places for all log values also for later.

Line 458: The objective of the study was to examine instead of The study sought to examine

The conclusion is largely a summary (a repetition of the results) rather than a true conclusion. Please correct.

Comments on the Quality of English Language

The language of the article is mostly correct, but there are some mistakes, so it needs to be corrected in some places (which I have tried to list in my comments). 

Author Response

Response to Reviewer 3 Comments:

Comment on the manuscript “Antimicrobial activity of phloretin against Vibrio parahemolyticus and its application in seafood” by Siyang Chen, Wenxiu Zhu, Yiqun Zhan and Xiaodong Xia

The manuscript “Antimicrobial activity of phloretin against Vibrio parahemolyticus and its application in seafoods” by Siyang Chen and co-workers deals with an important topic. However, there are problems with the manuscript. The objective of this study was to examine the deeper understanding of the antibacterial mechanism of phloretin against V. parahemolyticus. The article's principal strength lies in the fact that the antimicrobial substance in question has not previously been subjected to such comprehensive study.

General comments:

The weakest part of the manuscript is the chapter on Materials and methods, which is written with many gaps. The work does not ensure reproducibility.

---- Thanks for your comment. We have added some specific details in the chapter on Materials and methods in the revised manuscript.

Specific comments:

Line 18: beeing instead of bening

---- Thanks for pointing this out. We have corrected “bening applied” to “being applied.”

Line 19: The colony counting method is not accurate enough to give the result to two decimal places.

---- Thanks. We now reported the colony count using numbers with one decimal place.

Line 29: Gram-negative instead of gram-negative

---- Thanks for pointing this out. We have corrected “gram-negative” to “Gram-negative” in the revised manuscript.

Line 36: „to avoid or remove „ - How can the contamination of V. parahemolyticus be removed from a food? I suggest to delete „or removed”.

---- Thanks. We have deleted “or removed”.

Line 41: Please specify, which tea extract it is.

---- Thank. It is a green tea extract. We have added it in the revised manuscript with its reference.

Line 65: of cell instead of ofcell

---- Thanks. We have changed “ofcell” to “of cell”.

Line 74: Please provide producer, country and product number.

---- Thanks. We have added TSB producers and countries to the revised manuscript.

Lines 76-77: „Other chemical agents used in 76 this study were of analytical reagent grade. „Please provide producer and country.

---- Thanks. We have added all chemical reagent producers and countries to the revised manuscript.

Line 77: Please specify the type of seafoods here.

---- Thanks. We have added the type of seafood (sea bass, shrimp, and oysters) in the revised manuscript.

Line 82: Add the concentration of PHL in the initial solution.

---- Thanks for pointing this out. The original PHL concentration was 500 μg/mL. We also added it in the revised manuscript.

Lines 109-110: then washed three times with PBS solution. instead of after which they are washed on three occasions with a PBS solution.

---- Thanks. We have revised it to “then washed three times with PBS solution”.

Line 123: V. parahemolyticus (italic) instead of V. parahemolyticus

---- Thanks. We have corrected it to italics.

Line 127: Add the concentration of PHL.

---- Thanks. The concentration of PHL was 125 μg/mL. We also added it in the revised manuscript.

Line 139: Specify what kind of fluorescent enzyme labeling instrument was used.

---- Thanks. We have added the producer and country of the instrument in the revised manuscript.

Line 147-149: Specify the producer and country of the AKP assay kit and microplate detector.

---- Thanks. We have added the producer and country of the AKP assay kit and microplate detector in the revised manuscript.

Line 152: Add the concentration of PHL.

---- Thanks. The concentration of PHL was 125 μg/mL. We also added it in the revised manuscript.

Line 155: In what was it resuspended?

---- Thanks. The cultured bacterial suspension was centrifuged and the precipitate was resuspended with 0.9% saline solution.

Line 162: Provide producer and country for microplate detector.

---- Thanks. We have added the producer and country of microplate detector in the revised manuscript.

Line 163: V. parahemolyticus (italic) instead of V. parahemolyticus

---- Thanks. We have corrected it to italics in the revised manuscript.

Line 168: Specify what kind of fluorescence enzyme marker was used.

---- Thanks. We have added the producer and country of fluorescence enzyme marker in the  revised manuscript.

Line 171: Add the concentration of PHL.

---- Thanks. The concentration of PHL was 125 μg/mL.

Line 172: Please indicate the conditions under which the incubation period was 1 hour and the conditions under which it was 2 hours.

---- Thanks. We have added that incubation conditions for 1 and 2 hours are 37 °C and 180 rpm/min in the revised manuscript.

Line 173: Please specify what does INT present.

---- Thanks. We have added the full name of INT in 2.1. Materials and Reagents, and explained the role of INT in the revised section 3.8.

Line 174: V. parahemolyticus (italic) instead of V. parahemolyticus

---- Thanks. We have corrected it to italics in the revised manuscript.

Line 175: Specify the producer and country of the microplate detector.

---- Thanks. We have added the producer and country of the microplate detector in the revised manuscript.

Line 178: Please specify the producer and country and product number for the fluorescent probe. V. parahemolyticus (italic) instead of V. parahemolyticus

---- Thanks. We have added the producers and countries of fluorescent probes in 2.1. Materials and Reagents. We have corrected V. parahaemolyticus it to italics in the revised manuscript.

Line 181: Add the concentration of PHL.

---- Thanks. The concentration of PHL was 125 μg/mL.

Line 183: Specify what kind of fluorescent enzyme labeling instrument was used.

---- Thanks. We have added the producer and country of fluorescent enzyme labeling instrument in the revised manuscript.

Line 184: V. parahemolyticus (italic) instead of V. parahemolyticus

---- Thanks. We have corrected it to italics in the revised manuscript.

Line 188: the protein instead of The protein; add the concentration of PHL.

---- Thanks. We have corrected “The protein” to “the protein” in the revised manuscript. The concentration of PHL was 125 μg/mL.

Line 189: Gram instead of gram; please specify the producer and country and product number of the protein extraction kit.

---- Thanks. We have corrected “gram” to “Gram”. We have added the producer and country of  the protein extraction kit in the revised manuscript.

Lines 190-191: Please specify the producer and country and product number of bicinchoninic acid (BCA) test kit.

---- Thanks. We have added the producer and country of bicinchoninic acid (BCA) test kit in the  revised manuscript.

Line 204: V. parahemolyticus cells instead of V. parahemolyticuscells cells

---- Thanks. We have corrected V. parahemolyticuscells cells to V. parahemolyticus cells.

Lines 204-208: Please indicate at what initial microbial count was the pathogen inoculated?

---- Thanks. The initial microbial count of the inoculated pathogen was approximately 106 CFU/mL.

Line 208: Add the concentration of PHL.

----- Thanks. The concentration of PHL was 125 μg/mL.

Lines 211-212: was spread on TCBS agar plates, which were incubated  instead of  was applied to the surface. The TCBS solid plates were inoculated with a blotting rod, and the plates were incubated

----- Thanks. We have corrected it in the revised manuscript as suggested.

Line 213: The V. parahemolyticus colonies were counted  instead of  The V. parahemolyticus cells colonies visible were counted

----- Thanks. We have corrected it in the revised manuscript.

Line 222: “using PHL as an indicator bacterium” PHL is not a bacterium. Please reconsider.

----- Thanks. We have corrected “using PHL as an indicator bacterium in the V. parahemolyticus assay” to “using V. parahaemolyticus ATCC 17802 as an indicator bacterium”.

Line 228: 3.8 instead of 3.79

----- Thanks. We have corrected it in the revised manuscript.

Figure 1.: log10 CFU/ml instead of log CFU/ml

----- Thanks. We have corrected it in the revised manuscript.

Line 256: which can be can be postulated  instead of  which can be can be postulated

----- Thanks. We have corrected it in the revised manuscript.

Line 264: PHL caused the cell damage  instead of  The caused the cell damage

----- Thanks. We have corrected it in the revised manuscript.

Lines 266-269: The following sentences (or parts of them) are may have been inserted due to a copy error, please correct. A significantly lower number of red blood cells were identified in individuals within the control group. In contrast, the proportion of red blood cells in the PHL-treated group was observed to be higher.

----- Thanks for pointing this out. We have revised the sentence as follows: There was very few cells with red fluorescence in the control group. In contrast, the number of cells with red fluoresence in PHL-treated group was much higher.

Figure 3: The fluorescence microscope images of V. parahemolyticus are not visible, and there is only an empty black square in parts A and B. These figures should be replaced.

----- Thanks. We have replaced the microscope image.

Line 283: „all 260 absorbance values exhibited a notable increase” – this is not correct. The values measured at 260 nm exhibited a notable increase

---- Thanks. We have corrected it in the revised manuscript.

Line 284: This suggests  instead of  The suggests

---- Thanks. We have corrected it in the revised manuscript.

Line 312-313: This indicates instead of The indicates

---- Thanks. We have corrected it in the revised manuscript.

Lines 353-354: The following is not clear. „the bacteria were not detected. Bacteria were certainly present, possibly not alive/not culturable. Please correct.

---- Thanks. We have corrected “the bacteria were not detected” has been corrected to “no bacterial growth was observed”.

Line 359: In order to establish this, instead of To this end,

---- Thanks. We have corrected it in the revised manuscript.

Line 362: restriction, consistent   instaed of  restriction. Consistent

---- Thanks. We have corrected it in the revised manuscript.

Lines 383-384: In my opinion, quite poorly, the marked contrast between the control and PHL-treated groups mentioned in the text was observed. I don't know if it is possible to improve the quality of the gel image afterwards, but it is not conclusive.

---- Thanks for your comments. We have deleted the SDS-PAGE results as also suggested by the other reviewer.

Lines 387-388: “the protein bands in the SDS-PAGE transformed light to dark” Where could this be seen?

---- Thanks for your comments. We have deleted the SDS-PAGE results and the related description as also suggested by the other reviewer.

Line 389: The finding is inconsistent with the results reported by Chen et al. instead of The finding is at odds with that reported by Chen et al.

---- Thanks. We have corrected “at odds with that” to “inconsistent with the results”.

Line 432: were investigated instead of was investigated

---- Thanks. We have corrected “was” to “were” in the revised manuscript.

Line 439: a 2.5-log reduction  instead of  a 2.48-log reduction, use one decimal places for all log values also for later.

---- Thanks. We have corrected “2.48” to “2.5” in the revised manuscript. And we also applied this format to other log values in the manuscript.

Line 458: The objective of the study was to examine instead of The study sought to examine

The conclusion is largely a summary (a repetition of the results) rather than a true conclusion. Please correct. 

---- Thanks. We have revised the conclusing as follows: In summary, we demonstrated here that PHL impacted on the cell morphology, compromised cell integrity of V. parahaemolyticus, and decreased the protein, ATP, and DNA contents in V. parahaemolyticus. Moreover, PHL inhibited the activity of several key enzymes and metabolism in V. parahaemolyticus, and caused an elevation of the intracellular ROS concentration. Finally, PHL successfully reduced the number of V. parahaemolyticus cells in seafood (sea bass, shrimp, and oysters). These findings suggest that PHL could be potentially developed as a natural preservative to reduce contamination with V. parahaemolyticus in seafood, thereby broadening the range of application of PHL in food industry. Further research is necessitated to determine the appropriate concentration and its impact on organoleptic properties of foods prior to its application in real scenarios.

Comments on the Quality of English Language

The language of the article is mostly correct, but there are some mistakes, so it needs to be corrected in some places (which I have tried to list in my comments). 

---- Thank you very much for pointing it out. We have carefully checked for and revised language issues and had the revised manuscript proof-read by a native English speaker.  

Round 2

Reviewer 2 Report

Comments and Suggestions for Authors

The authors addressed all my concerns.

I have only one minor suggestion. Please improve Figure 2A, as almost nothing is visible in the pdf version received. 

Comments on the Quality of English Language

No comments

Author Response

Response to Reviewer 2 Comments:

I have only one minor suggestion. Please improve Figure 2A, as almost nothing is visible in the pdf version received.

 ---- Thanks for your comment. We have improved the microscope image.

Reviewer 3 Report

Comments and Suggestions for Authors

The manuscript “Antimicrobial activity of phloretin against Vibrio parahemolyticus and its application in seafoods” by Siyang Chen and co-workers deals with an important topic. The objective of this study was to examine the deeper understanding of the antibacterial mechanism of phloretin against V. parahemolyticus. The article's principal strength lies in the fact that the antimicrobial substance in question has not previously been subjected to such comprehensive study.

General comments:

The authors have made the suggested changes to the manuscript.

Specific comments:

Line 216: change Aanually to Annually

Figure 2A still shows black empty squares. The fluorescence microscope images of V. parahemolyticus are not visible. Although the arrows would show some cells, but they are not visible in the version I was able to download.

Author Response

Response to Reviewer 3 Comments:

The manuscript “Antimicrobial activity of phloretin against Vibrio parahemolyticus and its application in seafoods” by Siyang Chen and co-workers deals with an important topic. The objective of this study was to examine the deeper understanding of the antibacterial mechanism of phloretin against V. parahemolyticus. The article's principal strength lies in the fact that the antimicrobial substance in question has not previously been subjected to such comprehensive study.

General comments:

The authors have made the suggested changes to the manuscript.

Specific comments:

Line 216: change Aanually to Annually

---- Thanks for pointing this out. We have corrected “Aanually” to “Manually” in the revised manuscript.

Figure 2A still shows black empty squares. The fluorescence microscope images of V. parahemolyticus are not visible. Although the arrows would show some cells,but they are not visible in the version I was able to download

 ---- Thanks. We have improved the microscope image.
